# Deciphering the Dual Role of *Heligmosomoides polygyrus* Antigens in Macrophage Modulation and Breast Cancer Cell Growth

**DOI:** 10.3390/vetsci11020069

**Published:** 2024-02-03

**Authors:** Patryk Firmanty, Maria Doligalska, Magdalena Krol, Bartlomiej Taciak

**Affiliations:** 1Center of Cellular Immunotherapy, Warsaw University of Life Sciences, J. Ciszewskiego 8, b. 23, 02-786 Warsaw, Poland; pfirmanty36@gmail.com (P.F.); magdalena_krol@sggw.edu.pl (M.K.); 2Department of Parasitology, Faculty of Biology, University of Warsaw, Miecznikowa 1, 02-096 Warsaw, Poland; m.doligalska@uw.edu.pl

**Keywords:** parasitic nematodes, immune response modulation, macrophage activation, breast cancer cell proliferation

## Abstract

**Simple Summary:**

Our research focuses on how parasitic worms, specifically *Heligmosomoides polygyrus*, can change the body’s immune response. These parasites can lower the body’s defense mechanisms, but exactly how they do this is not fully understood. We believe that a type of immune cell, called macrophages, might be involved. Changes in these cells could potentially help tumors grow. In our study, we looked at how substances produced by *H. polygyrus* affect these immune cells and, in turn, how they influence the growth of breast cancer cells in the lab. We discovered that these substances from *H. polygyrus* increase the activity of certain genes in the immune cells. These genes are responsible for both promoting and reducing inflammation. Additionally, these substances change the surface features of the immune cells. Our findings show that these changes in immune cells caused by *H. polygyrus* could lead to increased growth of breast cancer cells in a laboratory setting. This research helps us understand more about how changes in the immune system can affect tumor growth, which is important for developing new cancer treatments.

**Abstract:**

In our study, we explored how parasitic nematodes, specifically *Heligmosomoides polygyrus*, influence the immune response, focusing on their potential role in tumor growth. The study aimed to understand the mechanisms by which these parasites modify immune cell activation, particularly in macrophages, and how this might create an environment conducive to tumor growth. Our methods involved analyzing the effects of *H. polygyrus* excretory-secretory antigens on macrophage activation and their subsequent impact on breast cancer cell lines EMT6 and 4T1. We observed that these antigens significantly increased the expression of genes associated with both pro-inflammatory and anti-inflammatory molecules, such as inducible nitric oxide synthase, TNF-α, (Tumor Necrosis Factor) Il-6 (Interleukin), and arginase. Additionally, we observed changes in the expression of macrophage surface receptors like CD11b, F4/80, and TLR4 (Toll-like receptor 4). Our findings indicate that the antigens from *H. polygyrus* markedly alter macrophage behavior and increase the proliferation of breast cancer cells in a laboratory setting. This study contributes to a deeper understanding of the complex interactions between parasitic infections and cancer development, highlighting the need for further research in this area to develop potential new strategies for cancer treatment.

## 1. Introduction

*Heligmosomoides polygyrus*, an intestinal nematode found predominantly in rodents, has garnered significant attention as a model organism, particularly for its pronounced immunosuppressive effects on host organisms [1,2]. This nematode secretes a diverse array of molecules, including antioxidants, proteases, protease inhibitors, and cytokine orthologs, intricately modulating the host’s immune response. These secretions are not only instrumental in ensuring the parasite’s survival within the host but also play a critical role in the process of tissue regeneration [3,4]. The impact of these molecules extends to the skewing of the immune response, particularly marked by an increased secretion of interleukins such IL-4, IL-5 and IL-13 by various immune cells, leading to a bias towards T-regulatory cell activation [5]. While the immunomodulatory effect of the nematode can, at times, be beneficial to the host, its secreted molecules can also manifest toxic properties, potentially leading to significant tissue damage [6].

One of the most significant effects of *H. polygyrus* infection is seen in the modulation of host immune cells, particularly macrophages. Macrophages are critical in regulating immune system responses, and their functional diversity is contingent upon the effector molecules that activate them [7]. They are traditionally categorized into two phenotypes: classically activated macrophages (CAM or M1), and alternatively activated macrophages (AAM or M2). The former are typically activated by bacterial lipopolysaccharides (LPS) and interferon-γ during type 1 immune responses, while the latter are predominantly activated by high levels of IL-4 secreted during parasitic infections [8]. Notably, *H. polygyrus* infection has been shown to induce alternatively activated macrophages, underscoring the nematode’s significant role in immune modulation [9]. Additionally, *H. polygyrus* releases exosome-like extracellular vesicles (EVs), which are known to suppress type 2 macrophage activation [10]. However, recent research posits that the traditional dichotomy of macrophage phenotypes may not fully encapsulate the phenotypic diversity and complexity of these cells [11]. In light of these findings, our study endeavors to precisely evaluate the impact of *H. polygyrus* somatic and excretory-secretory antigens on the expression of selected genes and surface markers in mouse bone marrow-derived macrophages (BMDM) in vitro to gain deeper insights into how these antigens influence macrophage polarization.

The role of macrophages extends beyond the realm of infection and immune regulation, particularly in the context of oncology. Macrophages are ubiquitously present within the tumor microenvironment and play a pivotal role in cancer progression [12]. One of their primary functions, phagocytosis, is often subverted by cancer cells, which secrete immunosuppressive factors to diminish the macrophages’ ability to combat tumor cells [13]. The concept of tumor-associated macrophages (TAMs) is particularly intriguing; these macrophages, primarily alternatively activated, are believed to contribute significantly to tumor growth and metastasis [14,15]. The presence of increased macrophage numbers within the tumor environment has been linked to enhanced tumor growth and metastasis, highlighting their critical role in cancer biology [16,17]. Furthermore, macrophages are known to secrete growth factors that are essential for tissue repair under normal conditions, but these substances can be exploited by cancer cells, leading to uncontrolled tumor growth [18].

Recent studies, including those by Jacob et al., have revealed that antigens from *Heligmosomoides polygyrus* can significantly impact the proliferation of colon cancer cells [19]. This finding aligns with the broader understanding that parasitic infections and the antigens they produce can modulate tumor growth, albeit with varying impacts depending on the type of cancer and the specific parasitic species involved [20,21]. The World Health Organization (WHO) has classified infections by three specific parasitic species (*Schistosoma haematobium*, *Clonorchis sinensis*, and *Opisthorchis viverrini*) as biological carcinogens, further emphasizing the significance of parasitic infections in cancer risk [22]. Our study aims to build upon this foundation by examining the effects of *H. polygyrus* antigens on the proliferation of two distinct breast cancer cell lines, EMT6 and 4T1. These cell lines were chosen due to their ability to mimic human breast cancer and their significant responsiveness to TGF-β (Transforming Growth Factor), an action that may be replicated by parasite-origin cytokine mimic TGF-β [23,24,25,26,27]. By investigating two different breast cancer cell lines, we aim to discern if there are varying responses between cancer types with differing metastatic potential and immunogenicity [28,29].

Furthermore, studies have demonstrated that alternatively activated macrophages play a significant role in the increased growth of 4T1 tumor cells in vivo [30]. Given that *H. polygyrus* infection has been observed to induce alternatively activated macrophages [31], our study posits that the antigens from *H. polygyrus* could alter macrophage activation in vitro and subsequently affect their antitumor activity. This research not only aims to elucidate the complex interactions between parasitic infections and cancer development but also seeks to provide novel insights into potential therapeutic strategies targeting the tumor microenvironment.

## 2. Materials and Methods

### 2.1. H. polygyrus Antigens

Adult worms were collected from mouse intestine 28 days after infection. The worms were intensively washed three times with PBS supplemented with penicillin (500 U/mL), streptomycin (500 µg/mL) and L-glutamine (2 mM) for 2 h in 37 °C. Then, 100 worms were cultured in 1000 µL of RPMI medium (Mediatech, Inc. Corning, Manassas, VA, USA) supplemented with penicillin (100 U/mL), streptomycin (100 µg/mL) and L-glutamine (2 mM) for 24 h in 37 °C, 5% CO_2_. After 24 h, the supernatant was discarded and fresh medium was added. The worms were further cultured for 5 days. Every day, supernatant containing nematode excretory/secretory products was collected and concentrated by centrifugal columns with a molecular weight cutoff of 3 kDa (Amicon, Beverly, MA, USA). The pooled concentrated supernatant was sterile filtered using a 0.22 μm filter. Protein concentration was measured by the Bradford Reagent (Sigma-Aldrich Co. St. Louis, MO, USA). The excretory/secretory antigen preparation was stored at −80 °C until used.

For somatic antigen preparation, the worms (about 500 male and female adults) were sonicated in 0·5 mL PBS (136 mM NaCl, 2.7 mM KCl, 8.1 mM Na_2_HPO_4_, 1.5 mM KH_2_PO_4_, pH 7.4) and centrifuged. The supernatant was collected and filtered sterile, and the protein concentration was measured.

The chromatogram profile of somatic antigen fractions obtained by molecular separation using high-pressure liquid chromatography and SDS-PAGE is provided in a previous study [32,33].

To measure bacterial endotoxin levels in *H. polygyrus* antigens, we used the Pierce LAL Chromogenic Endotoxin Quantitation Kit (Thermo Scientific, Rockford, IL, USA) and followed the manufacturer’s protocol. The lipopolysaccharides (LPS) concentration was determined in two separate solutions of somatic antigens and in the excretory-secretory antigen solution. The somatic antigens solution (SN) had an LPS concentration of 0.097 EU/mL. The other somatic antigens solution (SS) had an LPS concentration of 0.037 EU/mL, and the excretory-secretory antigens (HES) had an LPS concentration of 0.093 EU/mL.

### 2.2. Cell Culture

Bone marrow-derived macrophages (BMDM) were obtained by an authorized laboratory animal handler. Hematopoietic stem cells were collected from the bone marrow of 8-week-old BALB/C mice and plated at 2 × 10^6^ cells per bacterial culture plate. The cells were cultured in DMEM medium supplemented with L929 cell-conditioned medium (20%), BCS (10%), and 1% penicillin and streptomycin, under standard conditions (37 °C, 5% CO_2_) for 5 days. Before starting the experiment, the cells were detached from the culture plate by removing the medium, washing with PBS, and adding CellStripper solution (Corning, New York, NY, USA).

EMT6 and 4T1 (murine mammary carcinoma cells) were obtained from the American Type Culture Collection (ATCC) (reference CRL-2755 for EMT6, reference CRL-2539 for 4T1) and cultured under standard conditions (37 °C, 5% CO_2_) in DMEM medium supplemented with BCS (10%) and 1% penicillin and streptomycin. Regular mycoplasma testing was performed for all cell cultures.

### 2.3. MTT Assay

The effect of *H. polygyrus* antigens on macrophage viability was assessed using the MTT assay (Cell Proliferation Kit, Sigma-Aldrich). MTT is chemically 3-(4,5-dimethylthiazol-2-yl)-2,5-diphenyltetrazolium bromide. First, 1 × 10^5^ cells/well were seeded in 96-well adhesion plates and 200 µL of DMEM medium supplemented with medium conditioned with L929 cells (20%), BCS (10%), and penicillin and streptomycin (1%) was added to each well. The final concentration of two types of somatic antigen solutions (differing in LPS concentration) and secretory-excretory antigens was 10 µg/mL. In addition to the negative control without antigens, a control with LPS (0.1 EU/mL) was performed, which was the highest LPS concentration measured among the tested antigens. The cells were incubated for 24 and 48 h at 37 °C and 5% CO_2_. The viability of macrophages was assessed by measuring the absorbance at 570 nm using the Infinite Plate Reader 2000 (Tecan, Männedorf, Switzerland). The experiment was performed 3 times with 4 technical replicates (n = 12).

### 2.4. RT PCR

Macrophages were incubated with *H. polygyrus* antigens at a concentration of 10 µg/mL for 24 h in standard conditions (37 °C, 5% CO_2_). After incubation, the culture medium was removed, and the cells were lysed using Fenozol Plus solution (A&A Biotechnology, Gdansk, Poland) using the RNA Total RNA Mini Plus isolation kit. The cell suspensions were then transferred to tubes and stored at −21 °C until RNA isolation. The experiment was performed three times.

Total RNA was isolated from macrophages using the RNA Total RNA Mini Plus kit (A&A Biotechnology) according to the manufacturer’s protocol. cDNA was then synthesized using the FIREScript RT cDNA synthesis mix with oligo (dT) and random primers (Solis BioDyne, Tartu, Estonia). The qPCR was performed using the SYBR Select Master Mix kit (Thermo Fisher Scientific) and 96-well plates from Roche (Basel, Switzerland) on a LightCycler 96 thermal cycler.

The primers were purchased from Oligo (Warsaw, Poland) (sequences can be found in the Appendix A, Appendix A). In this study, primers for genes coding for the following molecules were used: Arg1 (Arginase), iNOS (Nitric Oxide Synthase), YM1, IL-4, IL-10, IL-6, TNF-α, CD206, and CCL2 (chemokine (C-C motif) ligand 2). PPIA (Peptidylprolyl isomerase A) was used as the reference gene.

### 2.5. Flow Cytometry Analysis

The effect of *H. polygyrus* antigens on the expression of selected macrophage surface markers was investigated by incubating macrophages with *H. polygyrus* antigens at a concentration of 10 µg/mL for 48 h in standard conditions (37 ° C, 5% CO_2_). Cells were seeded at 4 × 10^5^ per well in a 24-well plate and were maintained in DMEM medium with 5 ng/mL of M-CSF (Macrophage colony-stimulating factor), 3% BCS (Bovine Calf Serum), and 1% penicillin and streptomycin. Two control groups were also included in the experiment, each consisting of cells grown in 8 wells, cultured with the addition of LPS at a final concentration of 0.1 EU/mL and 0.2 EU/mL, respectively. After the 48 h incubation, the cells were stained with Zombie Aqua™ Fixable Viability Dye according to the manufacturer’s protocol. Then, the cells were divided into two groups and incubated with different antibody combinations for 30 min on ice: (1) anti-CD11b (FITC), anti-CD206 (PE), anti-TLR4 (APC), anti-TLR2 (VioBlue 450), and anti-F4/80 (APC-Cy7); and (2) anti-CD11b (FITC), anti-CD124 (PE), anti-CD64 (PE-Cy7), and anti-F4/80 (APC-Cy7). The experiment was performed in duplicate. The cells were analyzed using a BD FACSCanto™ II Flow Cytometer, and the data were processed with FlowJo software v. 10.9.

The effect of *H. polygyrus* antigens on the proliferation of EMT6 and 4T1 cancer cells and the antitumor activity of macrophages was explored by incubating these cells together with *H. polygyrus* antigens in 24-well plates for 48 h under standard conditions (37 °C, 5% CO_2_). *H. polygyrus* antigens were added to the culture medium to a final concentration of 10 µg/mL. The experiment was performed in triplicate wells, with each well containing a total of 2 × 10^5^ cells at the start. In addition to the co-culture group, two control groups were included, one with cancer cells only and the other with macrophages only, both cultured in the same conditions as the co-culture group. The cancer cells and macrophages in flow cytometry analysis were distinguished by staining cancer cells with CellTrace ™ Violet fluorescent dye according to the manufacturer’s protocol. Cell Trace™ positive cells were taken as cancer cells, and Cell Trace™ negative cells as macrophages. The experiment was performed in triplicate with EMT6 cells (n = 9) or duplicate with 4T1 cells (n = 6).

After 48 h of incubation, the culture medium was transferred to 5 mL tubes. Before analysis by flow cytometry, 10 µL of CountBright™ Absolute Counting Beads was added to each sample. The samples were then analyzed on a BD FACSCanto™ II Flow Cytometer and the data were analyzed using FlowJo software.

### 2.6. Ethics

The use of laboratory animals in the experiments was conducted in accordance with the Polish Act on the Protection of Animals Used for Scientific or Educational Purposes. The experiments were approved by the 1st Local Ethical Committee for Animal Experiments in Warsaw with the approval number 482/2017 dated 24 January 2018.

### 2.7. Statistics

Data in the graphs are presented as mean ± standard deviation. A Kruskal–Wallis test was used to compare the results of the MTT assay, gene and surface marker expression. The analysis of the results of cell counts in the samples collected by the cytometer was performed using the Mann–Whitney U tests. The statistical analysis was conducted using GraphPad Prism 9 software (GraphPad Software, San Diego, CA, USA). A *p*-value < 0.05 (indicated by an asterisk in the graphs) was considered statistically significant.

## 3. Results

### 3.1. Impact on Macrophage Viability

Our initial examination focused on the viability of macrophages when exposed to *H. polygyrus* antigens. Utilizing the MTT assay, we assessed the metabolic activity of macrophages treated with a concentration of 10 µg/mL of these antigens. Over periods of 24 and 48 h, we observed no significant reduction in macrophage viability. This outcome, as detailed in Figure 1, suggests that the selected concentration of *H. polygyrus* antigens is non-toxic to macrophages, thus providing a viable model for further investigation into the immune-modulatory effects of these antigens. Additionally, we confirmed that the LPS concentration present in the antigen solution was not a contributing factor to macrophage viability.

### 3.2. Gene Expression Changes in Macrophages

Advancing our investigation, we next explored the effects of *H. polygyrus* antigens on the genetic expression profile of macrophages. Our findings, as illustrated in Figure 2a–d, indicate a significant enhancement in the expression of genes associated with pro-inflammatory and anti-inflammatory activities following exposure to these antigens. Notably, the expression of arg1, a marker for alternative macrophage activation, showed a marked increase. Concurrently, inflammatory markers, such as CCL2, iNOS, and IL-6, also exhibited elevated expression levels. However, due to the large error bar of iNOS expression resulting from differences in responses between experiments, it should be interpreted cautiously. This dual modulation of gene expression suggests that *H. polygyrus* antigens have a complex effect on macrophage activation, triggering both pro-inflammatory and anti-inflammatory pathways. In contrast, genes typically associated with alternatively activated macrophages, such as IL-4, I-10, CD206, and Chil3, did not show a significant deviation from control levels, as detailed in the supplementary data (Appendix A).

### 3.3. Macrophage Surface Receptor Modulation

Further analysis focused on the modulation of macrophage surface receptors by *H. polygyrus* antigens. Our results, presented in Figure 3a–c, revealed an increase in the expression of CD11b, F4/80, and TLR2 receptors on macrophages after a 48 h culture with these antigens, but this increase was not significant. This finding suggests an antigen-driven alteration in macrophage surface phenotype, potentially reflecting a shift in their functional capabilities. However, the expression of TLR4, as shown in Figure 3d, remained unchanged compared with the control group. The limited number of repetitions for these experiments indicates that while these findings are suggestive, they should be interpreted cautiously and require further validation.

### 3.4. Effect on Breast Cancer Cell Proliferation

A significant aspect of our research was the examination of the influence of *H. polygyrus* antigens on the proliferation of breast cancer cell lines EMT6 and 4T1. This assessment was conducted both in isolation and in co-culture with macrophages. Our results showing how many cancer cells were present in the culture after 48 h of culture in relation to untreated cells, depicted in Figure 4a,b, suggest that the addition of *H. polygyrus* antigens to the culture medium can lead to an enhanced proliferation rate of both EMT6 and 4T1 breast cancer cells. This proliferative effect was noted in both single cultures of cancer cells and in co-culture with macrophages. Remarkably, the response to somatic antigens, especially in co-culture with EMT6 cells (Figure 4a) and in single culture with 4T1 cells (Figure 4b), showed statistically significant differences when compared with the control groups. However, variability was observed in the response to excretory-secretory antigens between different biological replicates, highlighting the complex nature of the interaction between these antigens and cancer cells.

### 3.5. Overview

In summary, our results provide a comprehensive overview of the multifaceted effects of *H. polygyrus* antigens on macrophages and breast cancer cells. The findings reveal significant modulation in macrophage viability, gene expression, and surface receptor expression, as well as a notable influence on the proliferation of breast cancer cells. These results contribute to a deeper understanding of the intricate interactions between parasitic infections, immune response modulation, and tumor cell dynamics, offering a foundation for further research in this field.

## 4. Discussion

Our study addresses a significant gap in understanding how macrophages, activated by parasitic antigens from *Heligmosomoides polygyrus (H. polygyrus)*, influence pathology beyond their infection-related roles. These cells, known for both proinflammatory and regulatory activities, demonstrated a marked increase in inflammatory gene expression, particularly in inducible nitric oxide synthase (iNOS) and CCL2, upon exposure to *H. polygyrus* antigens. Additionally, a significant increase in TLR2 expression was observed on macrophages, while TLR4 expression remained unchanged.

iNOS is crucial in immune regulation and the promotion of pro-inflammatory responses [34]. TLR signaling via the MyD88 pathway induces NF-kB translocation, promoting proinflammatory cytokine synthesis [35]. In contrast, TLR2 expression in *Schistosoma mansoni*, another helminth model, correlates with anti-inflammatory responses through Treg induction [36]. CCL2, a monocyte chemoattractant protein, is often activated by inflammatory stimuli [37].

The pro-inflammatory effects of *H. polygyrus* antigens challenge the previous hypothesis of their predominantly anti-inflammatory nature. However, the hypothesis is not entirely negated due to increased arg1 expression, a marker of alternatively activated macrophages, indicating diverse macrophage phenotypes. This diversity is common in in vitro studies with macrophages stimulated by bacterial endotoxins [38]. Further research revealed that bone marrow-derived macrophages exposed to *H. polygyrus* antigens exhibit both pro-inflammatory and regulatory activation [9,39]. Moreover, extracellular vesicles (EVs) from *H. polygyrus* modulate macrophage activation, leading to the downregulation of molecules associated with both type 1 and type 2 immune responses [10].

*H. polygyrus* excretory-secretory antigens also increase macrophage maturation markers like CD11b and F4/80. A rise in F4/80 expression may suggest the induction of immune tolerance [40]. Comparisons with studies on *H. polygyrus*-infected mouse macrophages, which often display an immunosuppressive phenotype influenced by anti-inflammatory cytokines from T-helper cells, highlight differences in macrophage behavior [41,42,43].

Our in vitro approach offers a cost-effective, reliable way to study these antigens and their fractions, enhancing our understanding of their mechanisms without extensive animal infections. This is crucial for advancing knowledge about *H. polygyrus* antigens.

Our findings also indicate that a 24 h stimulation (Figure 2) with *H. polygyrus* antigens leads to much more significant gene expression changes in macrophages compared with 48 h stimulation; thus, we showed only the changes in expression after 24 h. This is a notable observation given that LPS impacts gene expression within just 6 h [44]. This suggests a complex, time-dependent macrophage response to different stimuli, necessitating further investigation into various stimulation durations and their effects on macrophage activity and molecular synthesis.

The paradoxical role of parasitic antigens, particularly from Intestinal parasites like *H. polygyrus*, is highlighted by their association with positive outcomes in inflammatory diseases such as allergies [45], autoimmune diseases [46], obesity, and autism [47]. Ongoing research is exploring the therapeutic potential of helminth antigens in these diseases [48]. Our study contributes to this field by assessing the impact of *H. polygyrus* antigens on macrophage viability, indicating no cytotoxic effects, though a broader evaluation of their effects on various immune cells is needed for conclusive therapeutic insights.

In cancer research, this study underscores the need to consider the complex relationship between the immune system and cancer development. *H. polygyrus* antigens might negatively impact leukocyte immune surveillance despite potential benefits in treating autoimmune diseases. This concern arises as alternative macrophage activation by these antigens has been linked to increased cancer cell proliferation and metastasis, worsening cancer patient outcomes [16]. Our study found that macrophages stimulated with *H. polygyrus* antigens showed increased expression of certain M1 markers. However, the co-culture experiments with EMT6 cancer cells suggested that these macrophages did not effectively phagocytose cancer cells. In contrast, macrophages co-cultured with more malignant 4T1 cancer cells showed phagocytosis even without H. polygyrus antigens, indicating that these antigens do not significantly alter macrophage activation in vitro.

Both somatic and secretory-excretory antigens from *H. polygyrus* similarly affected cancer cell proliferation in co-cultures with macrophages, though secretory-excretory antigens had a more pronounced impact on gene expression and macrophage effector molecules. This suggests that both antigen types contain components stimulating breast cancer cell proliferation. Identifying these specific molecules, possibly including a TGF-β mimic (Hp-TGM) known to bind to mammalian TGF-β receptors [27], will likely require advanced techniques like high-performance liquid chromatography. Considering TGF-β’s role in tumor growth [49,50], the impact of Hp-TGM on cancer cells is a critical area for further research. Additionally, *H. polygyrus* infection in colitic mice has been linked to the upregulation of angiogenesis-related growth factors [51], suggesting that the parasite secretes compounds affecting cancer cell growth.

The addition of *H. polygyrus* antigens to culture medium increased cancer cell proliferation in both the co-culture and single culture setups, suggesting that macrophages might not play a significant role in this process. Furthermore, the phenotype of macrophages stimulated by these antigens does not align clearly with classical or alternative activation categories. This is due to an observed increase in the expression of both pro-inflammatory and anti-inflammatory cytokines. Previous studies have shown that alternatively activated macrophages can enhance the proliferation of 4T1 cells [30]. However, our experiments did not demonstrate this effect, suggesting that *H. polygyrus* antigens do not induce alternative activation of macrophages in vitro. We also noted differences in the proliferation of EMT6 and 4T1 cells following the addition of *H. polygyrus* antigens. 4T1 cells, known for their high metastatic potential, are widely used in metastatic breast cancer research. In our study, stimulation of 4T1 cells with *H. polygyrus* antigens led to an increase in their proliferation. Interestingly, this proliferation was inhibited by macrophages, whether stimulated with *H. polygyrus* antigens or not. This effect was not observed in EMT6 cells, which have a lower metastatic potential. Additionally, 4T1 tumor cells produce higher levels of inflammatory cytokines compared with the less metastatic EMT6 cells [28,29], suggesting a heightened response to *H. polygyrus* antigen stimulation. Further investigation is needed to understand the effects of *H. polygyrus* antigens on different cancer types, especially given the diverse roles of macrophages in tumor microenvironments. The use of EMT6 and 4T1 breast cancer cell lines in our study, common in preclinical models for breast cancer immunotherapy research, underscores the importance of macrophage infiltration in breast cancer, contrasting with less evident correlations in colon cancer [52]. This highlights the value of using in vitro models with different cell types to evaluate the potential efficacy of therapies impacting cancer cell proliferation and the complex interplay between the immune system and cancer.

## 5. Conclusions

In summary, our study provides valuable insights into the complex relationship between parasitic antigens, macrophage activation, and their potential implications in cancer biology. While the antigens from *H. polygyrus* present intriguing possibilities for therapeutic applications, their role in cancer progression underscores the need for a nuanced understanding of their effects on the immune system. Further research, particularly focusing on the specific bioactive compounds within these antigens and their broader impact on various types of cancer, is essential to harness their potential in clinical applications.

## Figures and Tables

**Figure 1 vetsci-11-00069-f001:**
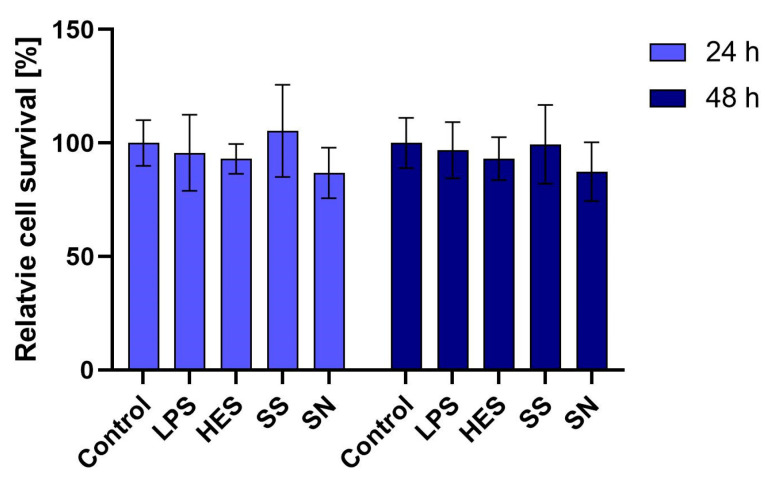
*H. polygyrus* antigens do not affect the viability of macrophages in vitro after 24 h or 48 h of culture. LPS—lipopolysaccharides, 0.1 EU/mL. Somatic antigens solution (SN) with an LPS concentration of 0.097 EU/mL, somatic antigens solution (SS) with an LPS concentration of 0.037 EU/mL and the excretory-secretory antigens (HES) with an LPS concentration of 0.093 EU/mL were added to the macrophages to obtain a concentration of protein of 10 µg/mL. Results were compared with the control (untreated macrophages). The experiment was performed 3 times with 4 technical replicates (n = 12). Error bars represent the standard deviation of the mean.

**Figure 2 vetsci-11-00069-f002:**
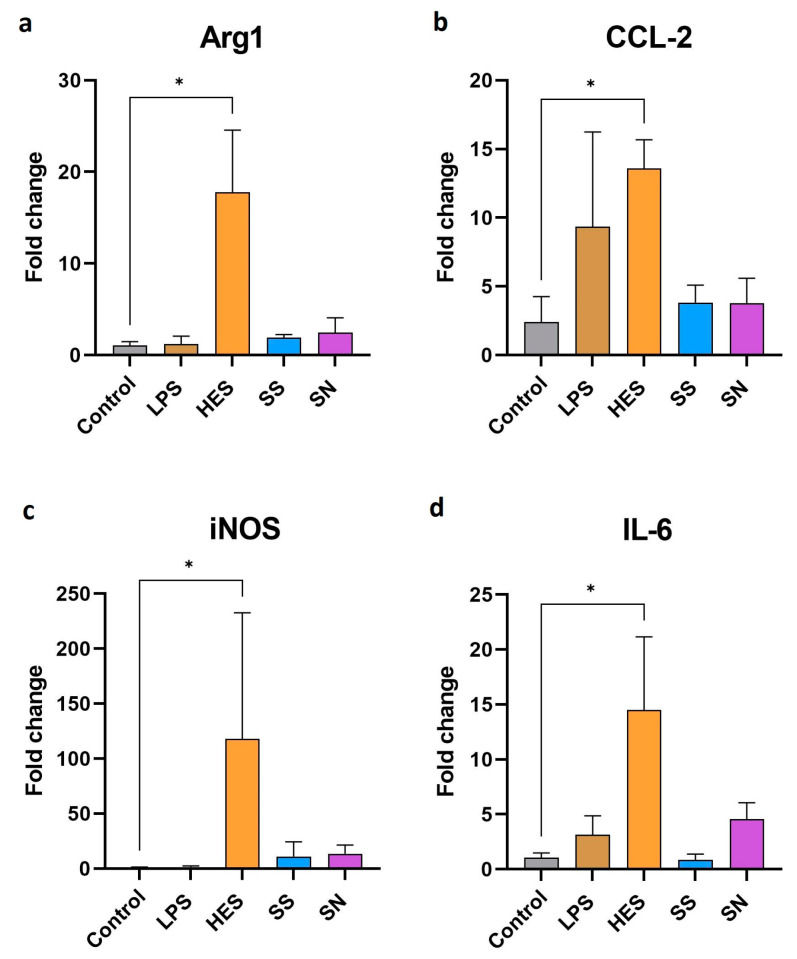
*H. polygyrus* excretory-secretory antigens increase the expression of genes encoding Arg1 (**a**), CCL2 (**b**), iNOS (**c**), and IL-6 (**d**) in macrophages after 24 h of culture in vitro. LPS—lipopolysaccharides, 0.1 EU/mL. Somatic antigens solution (SN) with an LPS concentration of 0.097 EU/mL, somatic antigens solution (SS) with an LPS concentration of 0.037 EU/mL and the excretory-secretory antigens (HES) with an LPS concentration of 0.093 EU/mL were added to the macrophages to obtain a concentration of protein of 10 µg/mL. Macrophages were cultured for 24 h. Results were compared with the control (untreated macrophages). The data are representative of three independent experiments. Significance was analyzed using GraphPad Prism, and a Kruskal–Wallis test was performed. Error bars represent the standard deviation of the mean. * *p* < 0.05.

**Figure 3 vetsci-11-00069-f003:**
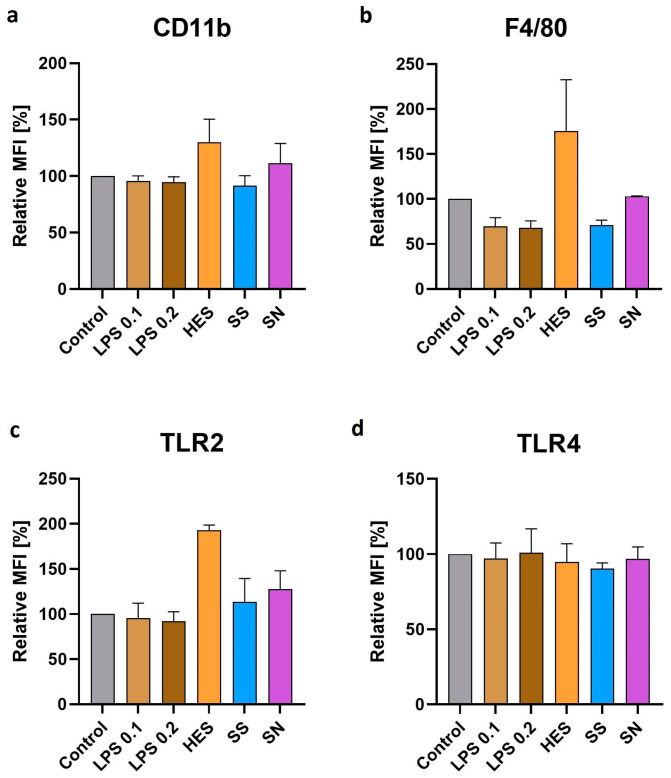
*H. polygyrus* excretory-secretory antigens increase the expression of CD11b (**a**), F4/80 (**b**), and TLR2 (**c**) in macrophages after 48 h of culture in vitro. TLR4 expression (**d**) did not show any changes. LPS 0.1—lipopolysaccharides at a concentration of 0.1 EU/mL, LPS 0.2—lipopolysaccharides at a concentration 0.2 EU/mL. Somatic antigens solution (SN) with an LPS concentration of 0.097 EU/mL, somatic antigens solution (SS) with an LPS concentration of 0.037 EU/mL, and the excretory-secretory antigens (HES) with an LPS concentration of 0.093 EU/mL were added to the macrophages to obtain a concentration of protein of 10 µg/mL. Macrophages were cultured for 48 h. Results were compared with the control (untreated macrophages). The markers expression differences were not statistically significant. The data are representative of two independent experiments. Significance was analyzed using GraphPad Prism, and a Kruskal–Wallis test was performed. Error bars represent the standard deviation of the mean.

**Figure 4 vetsci-11-00069-f004:**
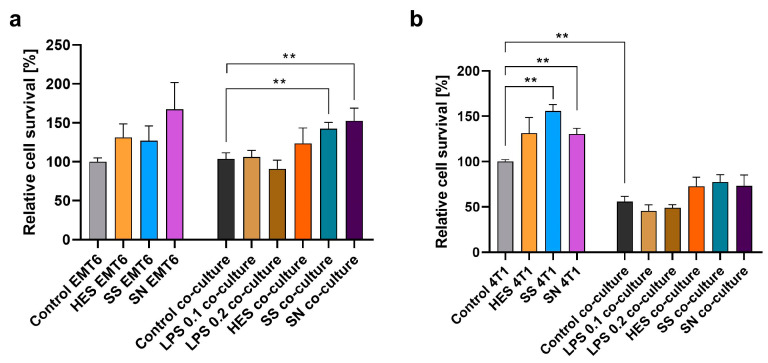
*H. polygyrus* antigens increase the proliferation of EMT6 (**a**) and 4T1 (**b**) breast cancer cells in vitro. LPS 0.1—lipopolysaccharides at a concentration of 0.1 EU/mL, LPS 0.2—lipopolysaccharides at a concentration 0.2 EU/mL. Somatic antigens solution (SN) with an LPS concentration of 0.097 EU/mL, somatic antigens solution (SS) with an LPS concentration of 0.037 EU/mL, and the excretory-secretory antigens (HES) with an LPS concentration of 0.093 EU/mL were added to the macrophages to obtain a concentration of protein of 10 µg/mL. Cells were cultured for 48 h. Results were compared with the control (untreated cancer cells or untreated cancer cells and macrophages). The data are representative of three (EMT6) or two (4T1) independent experiments. Significance was analyzed using GraphPad Prism and the Mann–Whitney U tests were performed. Error bars represent the standard deviation of the mean. ** *p* < 0.01.

## Data Availability

The raw data from this study are available upon request from the corresponding author.

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
