# Peer review of "Deciphering the Dual Role of Heligmosomoides polygyrus Antigens in Macrophage Modulation and Breast Cancer Cell Growth"

_vetsci, 2024, doi:10.3390/vetsci11020069_

Round 1

Reviewer 1 Report

Comments and Suggestions for Authors

This manuscript described on antigen positive or negative effects of H. polygyrus, an intestinal nematode found predominantly in rodents, on mouse macrophages. Although the manuscript included limited information based on in vitro, it is interesting for its role as antigens in parasites.

1. This work produced all results using parasite antigens. However, the simple expression that it is based on published papers is used for the composition and manufacturing method of the antigens. Therefore, since the antigen is a very important element in this paper, the preparation method, composition, and verification of the antigen(somatic antigens and H. polygyrus excretory-secretory) should be included.

2. Abbreviation needs the full names as separate section or in text.

3. Cell counts have been described as “The proliferation of 184 cancer cells was evaluated by measuring the fluorescence of CellTrace Violet dye” How did you distinguish between cancer cells and macrophages in co-culture to create figure 4?

4. In Figure 1, what is control and HESS? Add medium and the H. polygyrus excretory-secretory.

5. In figure 3, are there any significant differences between control and HES groups? Please add a notation indicating statistical differences.

6. In line 275, Figure 1 is Figure 4.

Author Response

Dear Reviewer,

We would like to express our sincere gratitude for the time and attention you have dedicated to reviewing our manuscript. Your insightful comments and suggestions have been invaluable in guiding us towards enhancing the quality and clarity of our work.

In response to your feedback, we have thoroughly reviewed and revised the manuscript. We have made every effort to address each of your comments and incorporate them appropriately. These revisions, we believe, have significantly improved our manuscript, ensuring it meets the high standards of publication.

Please find attached the revised version of the manuscript, along with a detailed response to each of your comments. We have strived to ensure that all your concerns have been adequately addressed.

We appreciate your guidance and are hopeful that the modifications we have made align with your expectations and the journal's standards.

Thank you once again for your valuable contribution to our work. We look forward to any further suggestions or comments you may have.

Reviewer 2 Report

Comments and Suggestions for Authors

Title: put Heligmosomoides in title and in abstract line 24.

lines 114-118: be useful to the reader if more information regarding preparation of antigens is provided and also measurement of bacterial endotoxin levels.  Do the levels of endotoxins vary between preparations?

Figure 1 and other legends: in legends add in full all abbreviations; also give n number of experiments in legends; the reader should be able to understand the figure from the legend without reference to the text.

Figure 2c: comment on large error bar.

Why did authors use 24hr for figure 2 and 48hr for figure 3 and 4?

line 245, add to end of sentence: but this increase is not significant.

lines 368/9: comment on this observation; line 269, after cells, add, 4b.

lines 270-273: in which case why did the authors not do more repeat experiments? 

line 275: this is figure 4.

In figure 4, relate relative cell survival to proliferation rate, pleased explain.

Why was 4T1 only done twice? 

Section 3.5: expand this section and provide data in support of comments or remove it.

lines 331-336: refer to relative figures in results section.

Minor points: line 198 and elsewhere: data are pleural; figure 4, y axis, relative.

Author Response

(The authors gave the same response as above.)

Round 2

Reviewer 1 Report

Comments and Suggestions for Authors

The revised paper needs English correction.

-in line 117, for 2 h in 37°C then 100 worms may be for 2 h in 37°C , and then 100 worms

-in line 136, check "and 135 followed the manufacturer’s protocol"

Comments on the Quality of English Language

Although the content is understandable, many sentences are worded incorrectly. Therefore, I think the English language needs to be checked.